# The Role of Nutritional Support for Cancer Patients in Palliative Care

**DOI:** 10.3390/nu13020306

**Published:** 2021-01-22

**Authors:** Paolo Cotogni, Silvia Stragliotto, Marta Ossola, Alessandro Collo, Sergio Riso

**Affiliations:** 1Department of Anesthesia, Pain Management and Palliative Care, Intensive Care and Emergency, Molinette Hospital, University of Turin, 10126 Turin, Italy; 2Medical Oncology 1, Veneto Institute of Oncology-IRCCS, 35128 Padova, Italy; silvia.stragliotto@iov.veneto.it; 3Clinical Nutrition and Dietetics Unit, Department of Internal Medicine, Molinette Hospital, 10126 Turin, Italy; mossola@cittadellasalute.to.it; 4Clinical Nutrition and Dietetics Unit, Maggiore della Carità Hospital, University of Eastern Piedmont, 28100 Novara, Italy; sergio.riso@maggioreosp.novara.it (S.R.); alessandro.collo@maggioreosp.novara.it (A.C.)

**Keywords:** oncology, nutritional status, nutritional support, artificial nutrition, home care, guidelines, clinical practice

## Abstract

The role of nutritional support for cancer patients in palliative care is still a controversial topic, in part because there is no consensus on the definition of a palliative care patient because of ambiguity in the common medical use of the adjective palliative. Nonetheless, guidelines recommend assessing nutritional deficiencies in all such patients because, regardless of whether they are still on anticancer treatments or not, malnutrition leads to low performance status, impaired quality of life (QoL), unplanned hospitalizations, and reduced survival. Because nutritional interventions tailored to individual needs may be beneficial, guidelines recommend that if oral food intake remains inadequate despite counseling and oral nutritional supplements, home enteral nutrition or, if this is not sufficient or feasible, home parenteral nutrition (supplemental or total) should be considered in suitable patients. The purpose of this narrative review is to identify in these cancer patients the area of overlapping between the two therapeutic approaches consisting of nutritional support and palliative care in light of the variables that determine its identification (guidelines, evidence, ethics, and law). However, nutritional support for cancer patients in palliative care may be more likely to contribute to improving their QoL when part of a comprehensive early palliative care approach.

## 1. Introduction

The role of nutritional support for cancer patients in palliative care is still a controversial topic. In the past, there has been limited collaboration between oncologists, clinical nutrition specialists, and palliative care physicians involved in the care of advanced cancer patients. Collaboration has been made more complex by the fact that, while it is clear what nutritional support is, there is no common or shared definition, not so much of palliative care, but of what constitutes a person who needs palliative care. In fact, there is no consensus in the literature on the definition of the palliative care patient because of the ambiguity in the common use in medicine of the adjective palliative [1]. For many years, efforts have been made to find a screening tool to identify patients in need of palliative care in the hospital setting [2], as this would be very useful in both Internal Medicine and, in particular, Medical Oncology [3].

Palliative care was established in the United Kingdom 50 years ago [4]. In this country, the General Medical Council defines people approaching the end of life (EoL) as those who are likely to die within the next 12 months [5]. This definition includes people with advanced, progressive, and incurable conditions. In fact, more than one-third of hospitalized cancer patients die or are transferred to hospice [6]. One survey reported that an unplanned hospitalization for a patient with advanced cancer strongly predicts a median survival of less than 6 months [7]. However, it is much more important to identify the needs rather than the exact prognosis of the palliative patient [5].

Timing is among the most important variables in identifying the indication for nutritional support in cancer patients in palliative care. A 2013 editorial indicated that palliative care is not an alternative at the end of curative treatments, but rather that they should be both simultaneous and early [8]. Therefore, all healthcare providers involved in the care of advanced cancer patients should be able to identify patients at risk of earlier death due to malnutrition rather than cancer.

The purpose of this narrative review is intended to be a brief guide to prescribing nutritional support based on the European Society for Clinical Nutrition and Metabolism (ESPEN) guidelines [9] and analysis of the literature evidence in palliative cancer patients. Specifically, the aim is to identify in the cancer patient the area of overlapping between the two therapeutic approaches, consisting of nutritional support and palliative care, in light of the variables that determine their identification (guidelines, evidence, ethics, and law) (Figure 1).

## 2. Cancer Patients in Palliative Care

According to Global Cancer Observatory (GLOBOCAN) estimates, about 18.1 million new cancer patients and 9.6 million cancer-related deaths occurred in 2018 worldwide. Many patients are now cured or are living longer with metastatic disease due to advances in diagnostics and treatments [10].

In recent years, patients with advanced cancer were defined as those with distant metastases, late-stage disease, and/or with prognosis of 6 to 24 months. Now, thanks to treatment advances, these patients live for multiple years, and many cancers are transforming into chronic diseases.

Palliative care patient have a neoplasm not responsive to curative treatment (World Health Organization-WHO, 1990) or a life-threatening disease (WHO, 2012). However, palliative care is not synonymous with EoL care or terminal care. By origin, the term “palliative” is derived from the Latin word “pallium” meaning “mask” or “cloak”. This etymology indicates what palliative care essentially is: cover or masking the symptoms and the effects of incurable disease for alleviating or reducing suffering [1,11,12].

Cancer is a systemic, complex, and heterogeneous disease. The cancer diagnosis, the disease itself, and the sequelae of cancer treatments are important stress factors for patients and their family. Cancer-related physical symptoms, together with psychological distress, social, and spiritual needs arising in the course of the disease, severely affect the patient’s and family’s life. Patients with advanced, incurable cancer often experience a symptom burden (including pain, dyspnea, fatigue, weight loss, and depression), emotional, social, existential, and spiritual suffering over the course of disease. Cancer symptoms depend on the stage, type of cancer, age, general condition of the patient, and many other factors. These symptoms impair the patient’s daily routine and quality of life (QoL) [13].

Moreover, in relation to the type of cancer, patients receive different types of treatments (chemotherapy, immunotherapy, radiotherapy, surgery, and other anticancer treatments) which lead to side effects, toxicities, and in some cases, permanent impairment resulting in disability. Symptom control is an essential part of cancer treatment, and more studies show positive effects of integrating palliative care early in oncology care to better address patients’ needs.

Palliative care is focused on symptoms and disease stress control for all cancer patients. The goal is to improve QoL for both the patient and the family, especially when disease-modifying interventions are not available. The WHO has proposed the following definition of palliative care: “Palliative care is an approach that improves the QoL of patients and their families facing the problems associated with life-threatening illness, through the prevention and relief of suffering by means of early identification and impeccable assessment and treatment of pain and other problems, physical, psychosocial and spiritual” [11,12].

A few systematic reviews concluded that early palliative care in patients with advanced cancer significantly improved patients’ QoL and could decrease symptom intensity. For this reason, oncology societies are committed to integrating palliative care into oncology.

The American Society of Clinical Oncology (ASCO) suggested the palliative care integrated early into oncology care is helpful for patients and families and complements the anticancer treatments [14,15,16,17]. The European Society of Medical Oncology (ESMO) proposed the term “patient-centered care” to defined care that aims to optimize the comfort, function, and social support of the patients and their families at all stages of the illness. To offer optimal patient-centered care for patients with advance cancer, the integration of supportive care and palliative care in oncology is necessary [18,19]. The Italian Association of Medical Oncology (AIOM), according to ESMO and ASCO programs, recommends an early integration of palliative care in cancer treatments. Additionally, due to the prevalence of severe and multiple symptoms, patients with advanced cancer can be referred to interdisciplinary palliative care teams [20,21].

## 3. Impact of Disease and Treatments on Nutritional Status in the Cancer Patient

Cachexia and anorexia have been invoked as “cancer’s covert killer” [22], and clinical data suggest that about 20–30% of deaths are attributable to malnutrition rather than cancer [23]. Malnutrition including muscle wasting, on the other hand, are recognized as common consequences of anticancer treatments. Whether these processes are reversible is a matter of debate, with the pathophysiological mechanisms involved being increasingly studied.

Cachexia is a multifactorial syndrome resulting from host factors, cancer type and stage, and treatment modalities. In pre-clinical stages, hormonal dysregulation and metabolic abnormalities occur as a result of the cancer microenvironment and chronic inflammatory state: insulin resistance, increased proteolytic activity, and lipolysis [24,25]. In later stages, a negative protein and energy balance derived from metabolic derangements results in progressive functional impairment with clinical manifestations characterized by hypophagia, early satiety, fatigue, and wasting.

Involuntary weight loss has been considered the hallmark of cachexia for at least 40 years and has been well recognized as an independent prognostic factor in cancer patients over the past 15 years [24,26]. In spite of the increasing prevalence of overweight and obesity in advanced cancer, ranging from 40% to 60% [27,28], it has been observed that almost 50% of patients are at nutritional risk and 13% are malnourished and have worse outcomes [29]. A grading system based on body mass index (BMI) and weight loss was proposed accordingly, comparing the impact on mortality of lower versus higher initial BMI: the highest risk category is patients with low initial BMI and high weight loss. These results show that a single cut-off of weight change for defining cachexia is misleading, since subgroups of patients with different degrees of risk can be defined [27].

It follows that changes in body weight are an imprecise means of appraising nutritional deterioration, whereas altered body composition is now acknowledged as the key feature to reveal the progression from malnutrition to cachexia. Important findings have been reported using magnetic resonance imaging and computed tomography to assess body composition in cancer patients. It has been noted that loss of muscle mass (sarcopenia) is particularly related to poorer tolerance to chemotherapy, increased risk of postoperative complications, deterioration of QoL, and survival [30,31,32,33]. Recent clinical literature suggests that intramuscular adipose tissue infiltration is another important and negative prognostic factor, as indicated by low muscle attenuation (radiodensity) on computed tomography [30,34,35]. Muscle steatosis, characterized by intramyocellular lipids, has been associated with poor muscle quality [36]. Moreover, visceral adiposity correlates with decreased treatment response and survival in many cancers and is associated with weight loss and muscle mass loss, as recently reviewed [37].

It has been noted that sarcopenia does not only result from cancer per se but can also be induced by chemotherapy. Iatrogenic sarcopenia is characterized by poor muscle quality which in turn results in a change in the volume of distribution of drugs, altered pharmacokinetics, and consequently increased toxicity. Sarcopenia has also been shown to correlate both with decreased response to chemotherapy and worse outcomes, in a troubling vicious circle [34,35].

The negative impact of sarcopenia on outcome also occurs in surgical patients, which is a risk factor for perioperative and postoperative complications [38,39,40]. Therefore, for frail patients with unresectable cancer, less invasive procedures are suitable in the elective setting, while palliative surgery is sometimes necessary for acute presentations, such as gastrointestinal obstruction, perforation, and bleeding. In recent years, conditions considered incurable such as peritoneal carcinomatosis have been more often subjected to cytoreductive surgery combined with hyperthermic intraoperative chemotherapy. Best practice for management comprises both pre-operative preconditioning with immunonutrition and postoperative early nutritional support [41].

Due to treatment-related side effects, surgery and chemotherapy cannot be undertaken as often as theoretically required for metastatic cancer patients, so radiation therapy (RT) is a commonly used alternative on their clinical journey [42]. Moreover, RT is often used in the palliative setting to obtain local cancer control and to mitigate symptoms. However, acute reactions to RT include mucositis, dysphagia, pain, vomiting, and diarrhea that may reduce adherence to treatment, necessitating aggressive nutritional intervention to enable patients to complete the course of treatment. Selection of eligible patients for palliative RT is challenging, as they may not benefit from treatment if their lifespan is too short to experience benefits from RT, or they may discontinue therapy early [43].

Head and neck cancers were associated with early discontinuation of palliative RT in a recent review; other predictive factors were low Karnofsky performance status (KPS) and long treatments courses [44]. Furthermore, for many patients with head and neck cancers, oral nutrition alone has been found to be inadequate to meet caloric requirements during courses of RT and/or chemoradiotherapy due to oral mucositis, making tube feeding necessary [45]. Gastrointestinal mucositis is a common consequence of fractionated abdominal irradiation, but it is probably under-reported due to being a clinical diagnosis. While its effects on food intake and absorption are well recognized, treatment and prevention strategies are limited, and clinical data on relative nutritional assessment are scant [46].

Whatever the etiopathogenesis, the functional decline in cancer patients is characterized by exacerbations of the disease in the last year of their life [47]. This clinical pattern was translated in a model of “catabolic crisis” with different clinical events related to disease progression or treatment, with a negative impact on nutritional status, with intercurrent phases of recovery in between crises. According to this model, no return to the previous functional baseline is observed during each recovery phase [48].

Recent clinical data seem to corroborate that “anabolic potential” can occur under certain conditions at defined phases of disease for advanced cancer patients [49]. Muscle gain has been related to stable disease, and it may represent response to successful cancer therapy. This highlights the importance of recognizing therapeutic windows for intervention before a refractory cachexia is established [50].

## 4. Dietary Counseling and Oral Nutrition Supplementation

In advanced cancer patients, preserving nutritional status may be a relevant concern during the palliative care phase. Even when the disease can no longer be cured, patients may survive for a reasonable amount of time (several months or years). In this context, nutritional status deficits may impair performance status, QoL, tolerance to palliative anticancer treatments, and survival. Therefore, patients with decreased oral intake require nutritional treatment in order to maintain nutritional status and meet the energy and protein needs [9].

Indeed, in the last phases of life, characterized by refractory cachexia with weight loss and deterioration of physical condition, nutritional care should be focused on recommending foods that the patient can tolerate and prefers to eat *(*“comfort feeding”), with the aim of ensuring a better QoL and alleviating symptoms [51].

The first goal of nutritional treatment is to preserve oral nutrition by minimizing food-related discomfort and maximizing food enjoyment through strategies including dietary counseling by a dietitian or other healthcare professionals, food fortification, and oral nutritional supplements (ONS) [52,53].

According to ESPEN guidelines, counseling is the first approach within a nutritional treatment, aimed at managing symptoms (appetite loss, nausea, early satiety, taste and smell changes, constipation, dysphagia, and psychosocial factors) and encouraging the intake of foods and drinks that are better tolerated, thus considering food intolerances and allergies, diet history, current meal pattern, and any changes in taste or smell that can affect preferences [9,54].

Dietary recommendations should be provided in order to optimize energetic and protein intake through modifications in food quality, size of portions, timing and splitting of meals throughout the day, and consistency adaptation. [51]

In this context, measures should be commensurate with the nutritional needs and predominant symptoms of each patient as part of a personalized and tailored nutritional treatment, such as that summarized in Table 1.

Moreover, patients should be made aware that healthy eating guidelines are no longer appropriate in their clinical conditions, and dietary restrictions should be avoided, as they limit food intake and enjoyment.

ONS find their use when nutritional requirements cannot be met by dietary counseling and food fortification. High-energy (>1.22 kcal/mL) and high-protein (>20% protein-derived energy) ONS allow the optimization of the caloric and protein supplies within a reduced volume, and special formulas could be advantageous in selected patients, such as semi-elemental products in malabsorption conditions [55]. According to a meta-analysis by Lee et al., the association of ONS administration and dietary advice seems to be more effective than ONS alone in relation to nutritional and functional outcomes (weight and fat-free mass gain/maintenance, QoL function score improvements) [56]. In the context of ONS, n-3 fatty acid-enriched formulas could provide some results in terms of weight gain and improvement in lean body mass, nutritional intake, and QoL [52,57]. However, such evidence appears to be limited by study heterogeneity in terms of cachexia stage, cancer site and stage, concomitant anticancer treatments, and endpoint measures [52].

## 5. Enteral Nutrition

Artificial nutrition (AN) can be integrated within a palliative care program when a positive influence on QoL is expected, and the risk of dying from malnutrition is higher than due to cancer progression [9,58]. ESPEN guidelines suggest that enteral nutrition (EN) should be first considered whenever the gastrointestinal tract is functional and oral nutrition remains inadequate despite nutritional interventions (counseling and ONS) [9].

EN is most frequently used in palliative care patients with head and neck or upper gastrointestinal cancers. In these patients, the primary indication for starting EN is oropharyngeal/esophageal dysphagia or gastric obstruction/dismotility, due to mechanical and functional factors related to the disease but also to palliative chemo- and/or radiotherapy induced side effects [59].

In a patient with a life expectancy of several weeks or months who is unable to fulfil more than 60% of their daily energetic needs in the long term through oral intake, it is a useful strategy to gain early gastrointestinal access. Among gastric devices, percutaneous endoscopic gastrostomy (PEG) is the gold standard, while radiologically inserted gastrostomy (RIG) or eventually surgical gastrostomy should be performed when an endoscopically guided tube cannot be placed. Long-term jejunal access (endoscopic or surgical jejunostomy) may be an option in the case of gastric obstruction/dismotility. Placement of a nasogastric tube (NGT) or nasojejunal tube (NJT) can be considered when short-term EN is expected (usually up to 6 weeks) and/or survival is uncertain [60].

In head and neck cancer patients who are unable to swallow, the use of an enteral route via NGT or gastrostomy may be a suitable strategy in order to achieve nutritional support in the setting of home care [59]. According to a recent study, evaluating the impact of home artificial nutrition (HAN) on performance status and survival in palliative cancer patients, EN, with dysphagia as the main indication, can maintain/improve the KPS and prolong mean survival up to 22.1 weeks (considering that death from starvation usually occurs within 2 months in healthy subjects, or even before in advanced cancer patients, without nutritional support) [61,62].

In esophageal cancer patients, PEG tends to grant a better nutritional status than self-expandable metal stent, and it is an independent factor associated with overall survival [63]. In these patients, endoscopically assisted NGT is also a feasible, low complication rate, palliative option for nutritional support, since it allows us to increase energy intake, serum albumin, median survival, and reduce hospitalization compared with nil per os [64]. However, Yu et al. indicate a slightly worse QoL in esophageal cancer patients using NGT feeding compared with the percutaneous route during chemoradiation therapy [65]. On comprehensive evaluation, it is reasonable to consider PEG as the preferred choice for long-term nutrition support in palliative esophageal cancer patients.

When EN is contraindicated or unfeasible, due to stenosis, sub-obstruction/obstruction, dysmotility, peritoneal carcinomatosis, malabsorption, abdominal pain, or intolerance and severe discomfort, parenteral nutrition (PN) should be considered [9].

Thus, in order to choose the optimal nutritional access, multidisciplinary clinical evaluation is strongly recommended, taking into account not only the primitive and secondary tumor locations (gastrointestinal vs. extragastrointestinal) and their direct/indirect effects on the digestive tract but also the patient’s overall clinical condition including cancer prognosis, nutritional status, performance status, QoL, potential effects of nutrition support, and the patient’s and his/her relatives’ wishes and expectations [9]. Table 2. summarizes the preferential nutritional routes in different cancer sites.

## 6. Parenteral Nutrition

Regarding nutritional support of patients with cancer, the ESPEN guidelines recommend “In a patient undergoing curative anticancer drug treatment, if oral food intake is inadequate despite counselling and ONS, supplemental enteral or, if this is not sufficient or possible, parenteral nutrition” [9]. However, when curative treatments are no longer available for unresectable locally advanced or metastatic disease, the goal of anticancer treatment is palliative [66]. In fact, chemotherapy is often intended as palliative therapy for patients with advanced cancer [67] because of the expected survival benefit [68]. In these patients, nutritional support should be offered and implemented considering the expected benefit on chemotherapy tolerance and consequently the potential benefit on survival [62].

In addition, ESPEN guidelines strongly recommend HAN, both enteral and parenteral, in cancer patients with persistent insufficient oral intake of nutrients or malabsorption in suitable patients [9].

With regard to the question of the route for delivering AN, this dispute is now over. Indeed, EN and PN are not competitors; conversely, EN and PN have specific indications and contraindications [69]. However, there are many factors that can negatively impact the delivery of EN in advanced cancer patients. Specifically, EN may not be able to meet nutritional needs in cancer patients with extensive bowel resections, high output ileostomy or intestinal fistula, as well as the presence of nutrition impact symptoms (nausea, vomiting, diarrhea, abdominal pain, and constipation due to peritoneal carcinomatosis). Orrevall et al. showed that nausea, vomiting, and gastrointestinal obstructions were the most common indications for PN in palliative patients [70].

A relevant question is the following: when is home parenteral nutrition (HPN) appropriate and suitable in patients without further anticancer treatments? Since 2009, ESPEN guidelines have stated that it is not a contraindication for HPN that oncologic treatment has been stopped [71]. For many years, clinicians have questioned whether all patients with advanced, incurable cancers should ever be sent home with PN [72]. ESPEN guidelines recommend proposing nutritional therapy in those not receiving anticancer treatments after considering, together with the patient, the prognosis and both the expected benefit on QoL and potential survival as well as the burden associated with HAN for them and their caregivers [9].

According to the classification of cancer cachexia [24], refractory cachexia is characterized by a low performance status (Eastern Cooperative Oncology Group-ECOG score 3 or 4) and a life expectancy of less than 3 months. At this stage of the disease trajectory, the cancer patient does not respond to anticancer treatments, just as he/she does not respond to AN aimed at reversing cachexia. This expert opinion is supported by data reporting that cancer patients within 90 days of death have a low probability that nutritional intervention will be able to stop or reserve cachexia [19].

Concerning the ethical aspect of this choice, there has been much debate about whether or not to feed the palliative cancer patient [73]. Denial of this treatment option elicits the following question, “Does this mean I’m going to starve to death?” [74]. This ethical dilemma represents a controversial issue. Indeed, despite the limited benefits, providing AN to cancer patients who are in their last weeks of life is a frequent practice [75].

Referring to the principles of bioethics and ESPEN guidelines [9], the prescription of HAN should be discussed with the patient respecting his/her autonomy and, as also required by law, his/her choice or advance directive to refuse AN.

Regarding clinical appropriateness, HPN is not recommended in patients with worsening clinical conditions (severe organ dysfunction or uncontrolled symptoms), low KPS (<50) or poor ECOG score (≥3), short estimated life expectancy, and patient refusal [69].

Prognosis is obviously an important conditioning issue. ESPEN guidelines have recommended that PN should be considered if the expected survival of cancer patient is greater than 2–3 months [76]. Indeed, predicting survival in incurable cancer patients is not easy, and validated scoring systems should be used [77].

Cancer patients on HPN are not all the same depending on whether or not they are receiving chemotherapy and receiving supplemental or total PN. In a large prospective study analyzing the clinical characteristics and predictive factors of survival of adult cancer patients receiving HPN, it was found that incurable patients receiving total PN were more frequently severely malnourished, more frequently had KPS <70, and had a higher grade of inflammation [78]. As a result, a four-fold lower survival was observed in these patients compared with in the cohort receiving chemotherapy and supplemental PN.

For palliative cancer patients receiving HPN, one of the most important elements that should be monitored is the need to adjust the prescription of HPN as well as when to wean off or discontinue this therapy [79].

Change in QoL represents another crucial issue in patients with cancer receiving HPN. Clinicians should identify cancer patients who might benefit from HPN and balance the potential advantages with HPN without prolonging life in those with no chance of improvement [80]. In a longitudinal study in advanced cancer patients, even those not receiving anticancer treatment, HPN was shown to significantly improve global QoL, physical, role, and emotional functioning [81]. Conversely, several studies identified that worsened QoL was associated with receipt of palliative chemotherapy in patients with advanced cancer at the EoL [66,82].

Sometimes, the use of HPN in cancer patients has been ruled out by clinicians concerned about the risk of complications (bloodstream infection, venous thrombosis, and catheter-related mechanical complications) from handling the central venous access device for HPN infusion. As a matter of fact, if carefully managed, HPN can be safely provided even to advanced cancer patients recording a low rate of complications [83].

Finally, an important question is whether there is evidence of a potential survival benefit depending on HPN in incurable cancer patients. However, there are two methodological difficulties in carrying out a controlled trial in malnourished cancer patients. First, it is not ethically acceptable to have a control group of patients with chronic insufficient food intake (aphagic or severely hypophagic) who receive no nutritional support. Second, in patients receiving both chemotherapy and HPN, there is no evidence of a possible survival advantage exclusively induced by HPN.

In a recent study, the authors attempted to eliminate these two methodological biases. The aim of this study was to compare the survival of malnourished cancer patients in palliative care, eligible for HPN according to guideline recommendations, who received HPN with a homogenous group of patients, equally eligible for HPN, who did not receive HPN but artificial hydration (AH) for logistic reasons or due to patient refusal [84]. Survival of the two groups showed a statistically significant difference favoring patients on HPN who had a median overall survival three times higher than that of the cohort who received AH (4.3 versus 1.5 months, respectively). This increase in median survival (2.8 months) is the same (2.7 months) as that found in cancer patients in the group receiving early palliative care compared with those in the standard care group [17].

## 7. Nutritional Support and Burden for Patients and Families

Nutrition impact symptoms are frequently encountered in patients with cachexia and are associated with adverse outcomes such as reduced intake, weight loss, and decreased survival [48,52]. However, trying to make a patient eat, when he/she has marked appetite loss, can lead to increased patient distress regarding interactions with his/her relatives/caregivers. Some authors reported stories of patients, in their dying days, pretending to be asleep when relatives visit, so that the relatives do not try to make them eat something.

The anxiety that surrounds eating and the related psychological distress causes negative impact on QoL of patients and their relatives [85]. The ESPEN guidelines recommend offering and implementing nutritional interventions in advanced cancer patients only after considering, together with the patient, the burden associated with nutritional support [9]. Usually, investigations on HPN use in patients with cancer reported that people had a favorable perception of the impact of HPN on their QoL [86]. In some instances, QoL may be related to the inability to eat, rather than the dependence on HPN itself [87,88]. Orrevall et al. described the sense of relief and security of both patients and relatives when the nutritional requirements were met through HPN [89]. Patients with ovarian cancer on HPN experienced a burden of treatment that did not mitigate the benefits of HPN; in particular, in the interviews, they stated that the motivation to live outweighed the constraints imposed, and patients and relatives recognized HPN as a lifeline and were grateful for it [80].

Increasingly, people with cancer are required to pay a portion of their treatment costs through deductibles and coinsurance. However, there are no data available in the literature on the financial resources spent by families of cancer patients, such as medical and living expenses, to initiate and adhere to recommended nutritional treatments.

## 8. Artificial Nutrition and Law

On June 2015, the European Court of Human Rights at Strasbourg, affirmed the council’s judgment dated 24 June 2014 authorizing the withdrawal of AN and AH from Vincent Lambert, a French national, rejecting the case of his parents who had challenged the council’s judgment in the European Court [90].

In February 2016, the French government enacted the Claeys-Leonetti law that established the rights for EoL patients and obliged clinicians to abide by any advance directives regarding treatments. In particular, this law recognizes the wishes expressed by the patient and establishes his/her rights to continue, restrict, or discontinue medical treatments such as AN and AH [91].

In December 2017, the Italian Parliament approved law no. 219/2017, which set out new rules on informed consent requirements and advance directives in medical treatments [92]. This law allows patients to express their preferences regarding medical treatments; in particular, the option to refuse AN and AH is permitted.

Like in other western countries (e.g., Germany, United Kingdom, Austria, Belgium, France, Spain, and USA), this Italian law allows for the withholding and withdrawal of AN and AH, which are defined as medical treatments.

Moreover, discontinuing AH and AH is a moral imperative when their continuation would not result in any real benefit to the patient or may give rise to clinical complications, as well as when the patient is in the last hours/days of life, in which the body is no longer in any condition to assimilate the nutrients artificially provided.

## 9. Limitations of the Review

Some limitations should be acknowledged. First, we presented a narrative rather than a systematic review. Second, because of the lack of evidence on the role of nutritional support for cancer patients in palliative care, the decision to prescribe this therapy is more a source of debate. Finally, the prescribing process described is tailored to the characteristics and needs of these patients and is not applicable to other malnourished patients, such as those receiving nutritional therapy for curative anticancer treatments or for longer periods of time.

## 10. Conclusions

Whether or not they are still undergoing anticancer treatments, malnutrition leads to low performance status, impaired QoL, unplanned hospital admissions, and reduced survival in palliative care patients. Due to the fact that nutritional interventions adjusted to individual needs may be beneficial, guidelines recommend the assessment of nutritional deficiencies in all of these patients. Accordingly, if oral food intake remains inadequate despite skilled counseling and ONS, the guidelines recommend considering home EN or, if this is not sufficient or feasible, HPN (supplemental or total) should be considered in suitable patients.

HAN is not recommended for palliative cancer patients with a short life-expectancy (less than 2 months), whereas it should be administered if the person is likely to die earlier due to malnutrition rather than progression of the malignant disease. In addition, HAN should be withdrawn in the case of worsening clinical conditions (onset of severe organ dysfunction or uncontrolled symptoms), downgrading of performance status, estimated life expectancy of days, and patient will.

In conclusion, nutritional intervention in EoL care should be tailored to the patient’s needs and is primarily intended to support comfort and QoL. However, nutritional support for cancer patients in palliative care may be more likely to contribute to improving their QoL when part of a comprehensive early palliative care approach.

## Figures and Tables

**Figure 1 nutrients-13-00306-f001:**
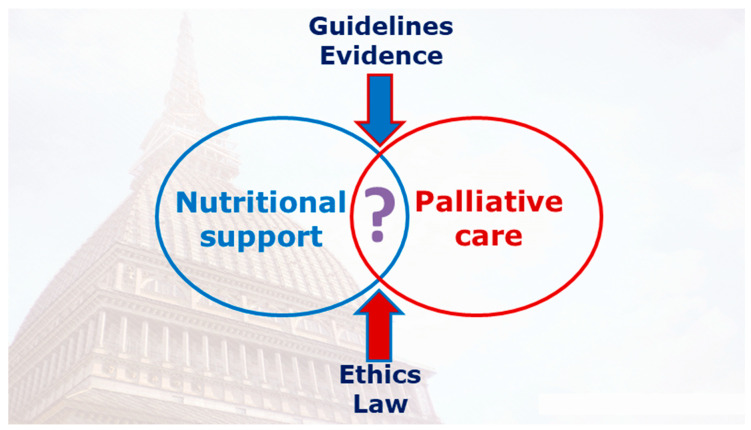
The area of overlapping between the two therapeutic approaches consisting of nutritional support and palliative care in light of the variables that determine their identification (guidelines, evidence, ethics, and law).

**Table 1 nutrients-13-00306-t001:** Dietary recommendations according to nutritional impact symptoms.

Symptoms	Dietary Recommendations
Appetite loss, anorexia	Minimize eating effort by preferring high energy and protein foods through small and frequent snacks throughout the day. High caloric liquid meals may be useful.
Taste and smell changes	Adjust diet in accordance with new taste preferences and by avoiding foods that may evoke aversion, such as those with an intense odor (roast meat, fish). Prefer mildly flavored foods. Cold foods are generally less odorous. If the oral mucosa is not sensitive, use salt, herbs, spices, and seasonings.
Nausea and vomiting	Prefer small and frequent snacks throughout the day (crackers, biscuits) in order to avoid stomach emptying. Take advantage of times when the patient feels less fatigued, or between cycles of chemotherapy. Less odorous and cold foods may be better tolerated.
Oral mucositis, pain	Prefer soft, creamy, or liquid foods, and avoid hard ones that could damage the oral membrane (nuts, hard fruit, crusts, hard baked goods). Prefer foods at room temperature, and avoid hot dishes and beverages. Ice cold foods and fluids may be pleasant. Avoid extreme tastes, such as spicy and acidic foods, citrus fruits, and very salty products.
Oropharyngeal dysphagia	Chopping or grinding and moisturizing food (adding cream, gravy, or sauce) allows an adequate thickness to be achieved to facilitate swallowing. Add a thickener to viscous foods in order to prevent choking. Avoid mixed consistency foods due to their high choking risk.
Esophageal dysphagia	Transit of bolus throughout the esophagus can be favored by chopping finely and dipping foods in liquids (drinks, gravy, or sauces). Chewing well and eating slowly and mindfully are recommended precautions, such as small and frequent meal consumption.
Constipation	An adequate liquid and fiber intake is aimed at preventing dehydration. Although 30–40 g of fibers per day is the goal for healthy subjects, this result is difficult to achieve in practice. Variate different types of fibers.

**Table 2 nutrients-13-00306-t002:** Preferential nutritional routes in different cancer sites.

Tumor Site	Preferential Nutritional Route	Comment
Head, neck	EN	Choose access according to the expected AN duration: short-term EN: NGT long-term EN: PEG (RIG or SG when endoscopic procedure is not feasible)
Chest: Esophagus, lung	EN	Choose access according to the expected AN duration: short-term EN: NGT long-term EN: PEG (RIG or SG when endoscopic procedure is not feasible) Self-expandable metal stents: lower survival benefit than PEG
Stomach	EN/PN	Choose access according to the expected AN duration: short-term EN: NJT long-term EN: PEJ (SJ when endoscopic procedure is not feasible)In presence of bowel sub-obstruction/obstruction, peritoneal carcinomatosis, severe gastrointestinal symptoms, or EN intolerance: consider PN
Pancreas, biliary tract, colon-rectum, uterus, ovary, bladder, prostate	PN	In presence of bowel sub-obstruction/obstruction, peritoneal carcinomatosis, severe gastrointestinal symptoms: consider PN
Others malignancies (e.g., brain, breast, blood)	EN/PN	Choose access according to the expected AN duration: short-term EN: NGT or NJT (if gastric dysmotility) long-term EN: PEG or PEJ (if gastric dysmotility) (RIG or SG or SJ when endoscopic procedure is unfeasible)In presence of bowel sub-obstruction/obstruction, peritoneal carcinomatosis, severe gastrointestinal symptoms, or EN intolerance: consider PN

Legend: AN: artificial nutrition; EN: enteral nutrition; PN: parenteral nutrition; NGT: nasogastric tube; NJT: nasojejunal tube; PEG: percutaneous endoscopic gastrostomy; PEJ: percutaneous endoscopic jejunostomy; RIG: radiologically inserted gastrostomy; SG: surgical gastrostomy; SJ: surgical jejunostomy.

## Data Availability

Not applicable.

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
