# Peer review of "The Role of Nutritional Support for Cancer Patients in Palliative Care"

_nutrients, 2021, doi:10.3390/nu13020306_

Round 1

Reviewer 1 Report

In the present review, the authors have reviewed the different aspects of nutritional support for cancer patients in palliative care. The topic is very relevant and the authors have well summarised the current practices and guidelines regarding nutritional support under different patient care conditions. As described in the review, it's equally important to have proper nutritional support to cancer patients undergoing different processes such as chemotherapy, surgery, etc, its high time to design and implement precision nutritional support to patients. Similarly, nutritional support for cachectic vs non-cachectic patients should be designed. It will be really good if authors can include a section of potential implications of individualized diet plans and discuss the different types of dietary support currently in practice and what kind of modifications may require to get the best patient care.

Author Response

Reviewer 1

It will be really good if authors can include a section of potential implications of individualized diet plans and discuss the different types of dietary support currently in practice and what kind of modifications may require to get the best patient care.

R. As suggested by the Reviewer, we have added a table (N. 1) reporting dietary recommendations according to nutritional impact symptoms.

Reviewer 2 Report

The authors present a narrative review regarding the nutritional support for cancer patients in palliative care. English editing by a native speaker is recommended before resubmitting it.

Title

・I feel it would be better to clearly state in the title that this paper is “a narrative review”.

Abstract

I think the authors should describe more about the finding from this paper in the abstract.

Text

・I feel that the authors need to reconsider the logical development in the “1. Introduction” section. In general, instead of writing the objective first, I think the authors should develop their logic in a direction that gradually clarifies the main points of their paper, starting from the broader background information toward the purpose of their study.

・The authors describe that “law” in Figure 1 is the important variable to determine the two therapeutic approaches, but I cannot find any discussion of specific law in the text. Please explain how the law influences the two therapeutic approaches.

・L82 Please make sure whether it is correct to have “complex” twice in one sentence.

・L118 “Malnutrition including wasting” rather than “Malnutrition and wasting” would be more suitable since WHO stated that wasting is one of the forms of malnutrition.

・I think it would be better to have a table summarizing the nutritional support for each type of cancer to clarify the points discussed in this paper (good and bad points, etc.).

・In considering the QoL of patients and their families, I also think the burden on professionals (caregivers, nutritionist, nurse, medical doctor, etc.) and families, who provide nutritional support, as well as financial resources such as medical and living expenses are important to discuss more in the text.

Author Response

Reviewer 2

English editing by a native speaker is recommended before resubmitting it.

R. A native English speaker has revised the manuscript to improve the English language.

Title

I feel it would be better to clearly state in the title that this paper is “a narrative review”.

R. As suggested by the Reviewer, we have used the words ‘narrative review’ in the title page, the Abstract, and the Introduction.

Abstract

I think the authors should describe more about the finding from this paper in the abstract.

R. As suggested by the Reviewer, we have revised the Abstract describing more about the finding from this review.

Text

I feel that the authors need to reconsider the logical development in the “1. Introduction” section. In general, instead of writing the objective first, I think the authors should develop their logic in a direction that gradually clarifies the main points of their paper, starting from the broader background information toward the purpose of their study.

R. According to the suggestions of the Reviewer, we have revised the Introduction.

The authors describe that “law” in Figure 1 is the important variable to determine the two therapeutic approaches, but I cannot find any discussion of specific law in the text. Please explain how the law influences the two therapeutic approaches.

R. As suggested by the Reviewer, we have added a new paragraph “Artificial Nutrition and Law” (N. 8).

L82 Please make sure whether it is correct to have “complex” twice in one sentence.

R. We have corrected the sentence.

L118 “Malnutrition including wasting” rather than “Malnutrition and wasting” would be more suitable since WHO stated that wasting is one of the forms of malnutrition.

R. We have corrected the sentence.

I think it would be better to have a table summarizing the nutritional support for each type of cancer to clarify the points discussed in this paper (good and bad points, etc.).

R. As suggested by the Reviewer, we have added a table (N. 2) summarizing preferential nutritional routes in different cancer sites.

In considering the QoL of patients and their families, I also think the burden on professionals (caregivers, nutritionist, nurse, medical doctor, etc.) and families, who provide nutritional support, as well as financial resources such as medical and living expenses are important to discuss more in the text.

R. As suggested by the Reviewer, we have added a new paragraph “Nutritional Support and Burden for Patients and Families” (N. 7).

Round 2

Reviewer 2 Report

The authors have addressed most of the comments and the revised manuscript has been much improved and is in a nice condition now.

English editing in terms of paragraph writing is recommended because there are too many paragraphs with a few sentences in the whole manuscript.